# Evaluation of Surgical Indications for Full Endoscopic Discectomy at Lumbosacral Disc Levels Using Three-Dimensional Magnetic Resonance/Computed Tomography Fusion Images Created with Artificial Intelligence

**DOI:** 10.3390/medicina59050860

**Published:** 2023-04-28

**Authors:** Katsuhisa Yamada, Ken Nagahama, Yuichiro Abe, Yoshinori Hyugaji, Daisuke Ukeba, Tsutomu Endo, Takashi Ohnishi, Katsuro Ura, Hideki Sudo, Norimasa Iwasaki, Masahiko Takahata

**Affiliations:** 1Department of Orthopaedic Surgery, Hokkaido University Hospital, Sapporo 060-8638, Hokkaido, Japan; yka2q@pop.med.hokudai.ac.jp (K.Y.);; 2Department of Orthopaedic Surgery, Sapporo Endoscopic Spine Surgery Clinic, North-16, East-16, Higashi-ku, Sapporo 065-0016, Hokkaido, Japan; 3Department of Orthopaedic Surgery, Eniwa Hospital, Eniwa 061-1449, Hokkaido, Japan

**Keywords:** full endoscopic lumbar discectomy, transforaminal approach, 3D MRI/CT fusion imaging, endoscopic surgery simulation, L5–S1 lumbar disc herniation, artificial intelligence

## Abstract

*Background and Objectives:* Although full endoscopic lumbar discectomy with the transforaminal approach (FED-TF) is a minimally invasive spinal surgery for lumbar disc herniation, the lumbosacral levels present anatomical challenges when performing FED-TF surgery due to the presence of the iliac bone. *Materials and Methods:* In this study, we simulated whether FED-TF surgery could be safely performed on a total of 52 consecutive cases with L5–S1 or L5–L6 disc herniation using fused three-dimensional (3D) images of the lumbar nerve root on magnetic resonance imaging (MRI) created with artificial intelligence and of the lumbosacral spine and iliac on computed tomography (CT) images. *Results:* Thirteen of the fifty-two cases were deemed operable according to simulated FED-TF surgery without foraminoplasty using the 3D MRI/CT fusion images. All 13 cases underwent FED-TF surgery without neurological complications, and their clinical symptoms significantly improved. *Conclusions:* Three-dimensional simulation may allow for the assessment from multiple angles of the endoscope entry and path, as well as the insertion angle. FED-TF surgery simulation using 3D MRI/CT fusion images could be useful in determining the indications for full endoscopic surgery for lumbosacral disc herniation.

## 1. Introduction

Full endoscopic lumbar discectomy (FED) is a minimally invasive spinal surgery for lumbar disc herniation. FED reportedly results in an earlier return to daily life activities compared with open or microendoscopic discectomy [1,2,3,4,5]. FED with the transforaminal (TF) approach (FED-TF), which reaches the intervertebral disc space through Kambin’s triangle, is the conventional technique when performing FED [1,2,6]. Kambin’s triangle is an anatomical passageway that extends from the intervertebral foramen to the disc, exiting nerve root (ENR), and dural canal [7,8,9,10]. In the TF approach, an endoscopic cannula is inserted between the dural canal and ENR from the side of the superior articular process (SAP) to reach the disc; this procedure can cause nerve root damage [1,11,12,13]. Thus, it is critical to understand the morphology of the structures involved in a TF approach to prevent nerve root injury [10].

FED-TF surgery is usually performed at lumbar levels L1–L2 to L4–L5. Meanwhile, FED-TF surgery for lumbar disc herniation at the lumbosacral levels (L5–S1 or L5–L6) may present anatomical challenges [5,14,15,16,17]. As the iliac crest is an obstacle during FED-TF surgery at the L5–S1 level and even at the L4–L5 level, some patients may not undergo FED-TF due to a high iliac crest [5,14,15,17]. However, it is difficult to determine whether patients can undergo FED-TF surgery based on preoperative imaging alone.

We developed a software program that automatically extracts a three-dimensional (3D) lumbar nerve root image from magnetic resonance imaging (MRI) results and lumbar nerve volume data using artificial intelligence (AI) and evaluated the morphology of Kambin’s triangle in 3D according to the actual FED-TF approach using 3D MRI/computed tomography (CT) fusion images of the lumbar nerve root and spinal images [10]. These 3D lumbar nerve root/spinal images allow for the visual and quantitative assessment of the morphology of Kambin’s triangle [10]. However, L5–S1 was excluded because the TF approach was limited by the iliac bone, in addition to the SAP and ENR in the previous study [10].

Therefore, we hypothesized that lumbosacral 3D MRI/CT fusion imaging that includes the iliac crest could be used for preoperative simulation of FED-TF surgery for disc herniation at the L5–S1 level. The aim of this case-series study was to analyze the morphology at the L5–S1 level, including that of the iliac crest, during the TF approach using 3D MRI/CT fusion images and to evaluate the feasibility of FED-TF surgery for disc herniation using preoperative simulation with 3D MRI/CT fusion imaging.

## 2. Materials and Methods

### 2.1. Patients

The present study was conducted with approval from the relevant institutional review board, and written informed consent was obtained from all participants. The inclusion criteria were patients with a single level of intervertebral disc herniation of L5–S1 or L5–L6 who were deemed to have an indication for surgery. Patients with multi-level herniated discs and patients with spinal canal stenosis requiring decompression surgery were excluded. Overall, 52 patients (28 men and 24 women; mean age ± standard deviation, 42.1 ± 12.6 years) who underwent surgery for L5–S1 or L5–L6 disc herniation at our institution between October 2020 and March 2022 were included in the study (Table 1). All patients were diagnosed with lumbar disc herniation via MRI and had clinical symptoms of lumbar disc herniation such as leg pain and/or low back pain. Standing posteroanterior and lateral radiographs of the whole spine were taken in all cases to confirm the presence of lumbarization and whether the herniated disc level was L5–S1 or L5–L6. The indication for surgery included persistent leg pain and neurological symptoms at the level of the herniated disc that were refractory to conservative therapy, including rest and medication. Axial and sagittal MR images were used to classify the herniation types (protrusion, subligamentous extrusion, transligamentous extrusion, or sequestration) [18,19] and the zone of herniated disc (central, subarticular, foraminal, or extraforaminal) (Table 1) [20].

### 2.2. 3D Fusion Imaging of Lumbosacral Nerve Root-Intervertebral Disc–Spine-Ilium

All patients underwent preoperative lumbosacral spine MRI and CT to create 3D fusion images of the lumbosacral nerve root-intervertebral disc–spine-ilium, and the images were used to simulate FED-TF as described below. The 3D fusion images were created using a workstation (SYNAPSE VINCENT, Fujifilm Co., Ltd., Tokyo, Japan) as previously reported [10]. MR images of the lumbosacral spine were obtained using a 1.5-T MR imaging system (Signa HD23; General Electric Healthcare, Chicago, IL, USA) with a 3D multiple echo recombined gradient echo sequence [10]. Three-dimensional lumbosacral nerve root MR images were created using AI software that automatically extracts three-dimensional lumbosacral nerve root images from MRI volume data using a deep neural network [10]. The software we developed can acquire 3D neuroimages in a very short time (within 10 s) with very simple commands, whereas this takes several hours with conventional manual methods [10]. Volume rendering images of the intervertebral disc and herniated disc manually created from MRI volume data were added to the automatically created 3D lumbosacral nerve image to create a 3D image of the lumbosacral nerve root and intervertebral disc (Figure 1). CT images of the lumbosacral spine and ilium were obtained using a 16-channel multi-slice CT scanner (Bright Speed, GE Healthcare, Chicago, IL, USA) [10]. Images were obtained in the supine position with a custom-made trunk support pillow to maintain the same position during MR and CT imaging [10]. The 3D fusion images of lumbosacral nerve root-intervertebral disc–spine-ilium were created by merging the 3D MR images of the nerve root and herniated disc and 3D CT images of the lumbosacral spine and ilium [10] (Figure 1).

### 2.3. FED-TF Simulation Using 3D Fusion Imaging at L5–S1

FED-TF simulation at L5–S1 or L5–L6 was performed using the fusion images of the 3D MR images of the lumbosacral nerve root and herniated disc and 3D CT images of the lumbar spine and iliac bone according to the actual surgical approach (Figure 2). First, the 3D fusion image was rotated along the axial plane of the L5 lower endplate to the approach side up to the angle where Kambin’s triangle maximally appears (Figure 2a). Next, the fusion image was then rotated cephalad so that the iliac crest did not interfere with Kambin’s triangle and to check if the herniated disc could be reached (Figure 2b,c).

A 3 mm diameter endoscopic virtual axis was placed according to the FED-TF approach (Figure 2d), and the following criteria were used to determine whether endoscopic discectomy could be performed: (1) whether the endoscopic virtual axis to the intervertebral disc could be approached without requiring osteotomy of the superior articular process, iliac crest, or transverse process; (2) whether it could reach the herniated disc or just underneath the herniated disc to resect the herniated disc; (3) whether it could be approached without contacting the nerve root; and (4) whether the pathway would not pass through the retroperitoneum (Figure 2e). Based on these criteria, two independent spine surgeons determined whether FED-TF surgery was feasible.

### 2.4. Surgical Procedure

A standard transforaminal full endoscopic lumbar discectomy (inside-out technique) was performed without partial osteotomy of the SAP (foraminoplasty) [6,11,21]. The endoscope was inserted using the puncture point and angle measured during the simulation. The arm of the fluoro machine was aligned with the axial direction of the endoscope, and the angle meter attached to the device was used to confirm that the arm angle was the same as in the simulation. Surgery was performed under local anesthesia with sedation used only enough to make the patient responsive to mild stimulation administered by anesthesiologists [22]. Electrodiagnostic testing systems are not used because it is possible to detect whether nerve roots have been stimulated during the surgery. The patient was positioned prone on a frame that allowed radioscopy. An 18-gauge spinal needle was inserted into the intervertebral disc from a position superior to the iliac crest and 9–13 cm from the midline [14]. After discography using a mixture of contrast media and indigo carmine, a guidewire was inserted into the disc, and a tapered cannulated obturator was inserted along the guide wire. A tubular working cannula (diameter, 7.9 mm) was then inserted into the disc along the obturator. An endoscope (Karl Storz, Tuttlingen, Germany) was inserted through the cannula, and the herniated disc and fibrotic scar tissues were resected as much as possible using endoscopic forceps and bipolar electrocautery (Elliquence, Baldwin, NY, USA). After the herniated fragment was removed, the endoscope was removed and sterile dressings were applied with sutures. No drain was placed. The patient was allowed to walk a few hours after surgery, and was discharged the next day.

### 2.5. Assessment

The feasibility of FED-TF surgery without foraminoplasty (partial resection of the SAP) was evaluated in cases that were judged to be operable based on the preoperative simulation. Clinical outcome evaluation of FED-TF surgery included the Japanese Orthopedic Association (JOA) score for lumbar spinal disorders (0–29, with higher scores representing a better status) and scores of low back pain and leg pain (visual analog scale [VAS]; 0–100 mm, with higher scores representing worse pain) preoperatively and at 1 day and 1 month postoperatively.

### 2.6. Data Analysis

All data are expressed as mean ± standard deviation. Paired-samples *t*-test was used to analyze pain scores before and after surgery. A *p*-value < 0.05 was defined as statistically significant.

## 3. Results

Based on the results of preoperative simulation, 13 patients with L5–S1 disc herniation (six men and seven women, mean age 40.5 years (range: 16 to 56 years) were judged to be operable based on the preoperative 3D imaging simulation. The 13 cases consisted of 3 cases of protrusion-type and 10 cases of subligamentous extrusion-type herniation, and the location of the herniated disc was in the central canal zone in 2 cases, subarticular zone in 9 cases, and foraminal zone in 2 cases.

All 13 patients underwent FED-TF surgery for L5–S1 disc herniations without foraminoplasty, and the surgical feasibility rate was 100%. None of the surgeries resulted in neurological complications. On clinical assessment, the mean JOA score significantly improved from 16.0 ± 4.5 preoperatively to 23.8 ± 4.2 immediately postoperatively and 25.2 ± 2.7 a month after surgery (*p* < 0.01, *p* < 0.01, respectively) (Figure 3). The mean VAS for low back pain significantly decreased from 60.0 ± 26.8 preoperatively to 17.0 ± 20.0 immediately postoperatively and 19.0 ± 15.1 a month after surgery (*p* = 0.01, *p* < 0.01, respectively). The mean preoperative VAS for leg pain was 71.3, which significantly improved to 26.3 ± 25.9 immediately after surgery and 20.0 ± 13.1 a month after surgery (*p* < 0.01, *p* = 0.03, respectively) (Figure 3).

### Case Presentation

A 28-year-old woman (case no. five) who had severe left leg pain (VAS: 80 mm) was diagnosed as having L5–S1 disc herniation (subligamentous extrusion type, subarticular zone) (Figure 4a). In the preoperative simulation, the lumbosacral 3D MRI/CT fusion image was rotated 65° to the right and tilted 5° cephalad with the lower endplate of the L5 vertebra serving as reference to maximally view Kambin’s triangle, from which the herniated disc (red) was confirmed (Figure 4b). The virtual axis of the endoscope (light blue bar, 3 mm in diameter) was advanced towards the herniated disc through Kambin’s triangle, and the axial fusion image confirmed that it did not pass through the retroperitoneum (Figure 4b). The patient was then judged to be suitable for FED-TF, and endoscopic discectomy was performed without foraminoplasty. The JOA score improved from 20 preoperatively to 28 postoperatively, and the leg pain VAS improved from 90 mm preoperatively to 0 mm postoperatively. MRI performed 1 week postoperatively confirmed that the herniated disc had reduced in size (Figure 4c).

## 4. Discussion

In the current study, 3D MRI/CT fusion images of the lumbosacral nerve root, intervertebral disc, lumbosacral spine, and ilium were used to simulate FED-TF surgery for L5–S1 disc herniation according to an actual spinal endoscopic surgical approach. FED-TF surgery was performed without causing neurological complications in all patients who were deemed operable according to the preoperative simulation, demonstrating the usefulness of this method.

In FED-TF surgery at the L5–S1 level, the iliac crest and the inclination of the L5–S1 level often obstructs the TF approach, resulting in a steep trajectory angle that reaches far from the herniated disc [14,23]. Furthermore, another concern is the large facet joint at the L5–S1 level, which also causes a steep trajectory angle [14]. For these reasons, FED-TF surgery for disc herniation at the L5–S1 level is less effective in some cases because the endoscope cannot reach the optimal location [16]. Therefore, it is important to evaluate the trajectory of the TF approach preoperatively to determine whether FED-TF can be performed, but this has been difficult to determine with preoperative imaging alone. Choi et al. [14] reported that FED-TF surgery at the L5–S1 level was possible without additional foraminoplasty when the height of the iliac crest was lower than the center of the L5 pedicle on lateral radiographs. Tezuka et al. [5] analyzed the entry trajectory into the L5–S1 disc using CT multiplanar images to evaluate the surgical feasibility of FED-TF surgery. The current study demonstrates that 3D MRI/CT fusion imaging can be used to visually check the trajectory during FED-TF surgery and to evaluate the feasibility of surgery. Because MRI/CT fusion imaging comprehensively evaluates the relationship between bones and nerves by capturing skeletal details on CT images and nerve details on MRI images [10], 3D simulation allows for easy understanding of the location of the nerve root, ilium, SAP, and herniated disc in the entry path (Figure 2).

Furthermore, this method can check the insertion angle and path of the endoscope from multiple angles and puncture points from the body surface (Figure 2). In the actual FED-TF surgery in this study, the endoscope was inserted using the puncture point and angle measured during the simulation. Intraoperative fluoroscopy was used to confirm the angle of puncture and to perform the surgery. All 13 patients who are deemed operable underwent FED-TF surgery with no neurological complications and clinical improvement, suggesting the usefulness of this FED-TF simulation.

The TF approach to the L5–S1 disc was a similar pathway in most cases, limited by the presence of the iliac crest. That is, the insertion path of the endoscope usually followed the superior border of the ilium to the anterior border of the superior articular process, reaching the intervertebral disc between the nerve root and the superior articular process. Therefore, the key to FED-TF surgery for L5–S1 level is to first apply the 18-gausge spinal needle to the superior articular process along the superior border of the ilium, and then along the inner border of the superior articular process to the intervertebral disc. FED-TF simulation with 3D MRI/CT fusion imaging allows visualization of the endoscopic insertion route in each case, which can be used in conjunction with intraoperative fluoroscopy to safely approach the intervertebral disc. In the future, we believe that this surgical simulation using 3D MRI/CT fusion imaging has the potential to be applied to the navigation system that can be used in real time during surgery.

In this study, a 3 mm diameter virtual axis was used for the FED-TF simulation. Although the actual endoscope diameter was 7.9 mm, surgery was possible without neurological complications in all cases in which the 3 mm diameter virtual axis reached the intervertebral disc. Insertion of the tapered cannulated obturator into the intervertebral disc while maintaining a good pathway with a guidewire would have increased the height of the disc and expanded the safety zone, thus allowing safe insertion of the endoscope. In this study, the diameter of the virtual axis was set to 3 mm with reference to the diameter of a high-speed drill, which is used to resect the superior articular process (foraminoplasty) in full-endoscopic surgery [24,25,26,27]. In other words, this simulation is intended for use in cases requiring foraminoplasty, and is expected to predict the width of the osteotomy of the superior articular process necessary to reach the herniated disc. We are considering applying this simulation to the outside-in technique of FED-TF surgery in the future [25,27,28].

The percentage of cases with L5–S1 level disc herniation that can be treated with FED-TF surgery remains unclear. Choi et al. [14]. reported that FED-TF surgery was performed on 100 cases of L5–S1 disc herniation; of these, foraminoplasty was not performed in 81 patients. Tezuka et al. [5]. analyzed the feasibility of FED-TF surgery at the L5–S1 level using the abdominal CT images of 323 cases and reported that the procedure could be performed in approximately 20% (right side: 24.1%, left side: 19.2%). In the current study, FED-TF surgery was performed without foraminoplasty in 13 of the 52 patients who underwent surgery for L5–S1 or L5–L6 disc herniation. The type of herniation in patients for whom FED-TF surgery could be performed was either protrusion or subligamentous types; there were no transligamentous or sequestration types, which are anatomically difficult to reach with the L5–S1 TF approach. Three-dimensional fusion imaging could be useful for preoperative planning because it enables surgeons to understand the positional relationship between the bone and nerve, including the herniated disc, even in cases not indicated for FED-TF surgery. Preoperative 3D simulation is a novel method to determine the indications for FED-TF surgery as well as other surgical procedures, including the FED interlaminar approach surgery [28].

In our previous study, we used 3D MRI/CT fusion images to analyze the morphology of Kambine’s triangle at L2–3, L3–L4, and L4–L5 levels [10], but excluded the L5–S1 level because we considered it a poor indication for total endoscopic surgery via the TF approach. However, this study revealed the pathway morphology of the TF approach to the L5–S1 level, indicating that this simulation can be applied not only to FED-TF surgery for herniation, but also to posterolateral full-endoscopic debridement and irrigation for pyogenic spondylodiscitis [22,29,30], discoblock, discography, and condoliase therapy for herniated disc [31].

This study has some limitations. First, CT and MR imaging were performed in the supine position, which is different from the supine position during surgery. The actual safety zone may be enlarged during surgery in the supine position compared with the preoperative simulation, as previously reported [10,11], and the indications for FED-TF surgery may be expanded beyond the indications in this simulation. Second, this study evaluated the feasibility of conventional FED-TF surgery for disc herniation at the L5–S1 level, but did not evaluate the feasibility of FED-TF surgery with foraminoplasty [5,6,14]. Foraminoplasty in addition to FED-TF can increase the zone of safety and pathway to the disc, expanding the indications for FED-TF surgery for lumbar herniation [5,6,14]. Preoperative simulation using 3D fusion imaging could be applied to FED-TF surgery with foraminoplasty to determine the margin of the SAP to be resected upon approaching the herniated disc. We did not compare the actual and the simulated surgical pathways in this study, which is another limitation of this study. Regarding this issue, we are analyzing the actual endoscopic insertion route from intraoperative fluoroscopic images in three dimensions and examining the accuracy of the preoperative simulation route in the future. The low number of cases is also a limitation.

## 5. Conclusions

We simulated FED-TF surgery using 3D MRI/CT fusion images of the lumbosacral nerve root, intervertebral disc, lumbosacral spine, and ilium. In the current study, FED-TF surgery was performed in all patients with L5–S1 disc herniation who were deemed operable by preoperative 3D simulation, indicating that the preoperative simulation using 3D MRI/CT fusion images is useful in preoperatively determining whether the endoscope can be inserted without nerve damage, while the path to the lumbosacral disc is limited by the presence of the iliac bone. Three-dimensional surgery simulation may properly assess the anatomical location of the endoscope path and its insertion angle from multiple angles. FED-TF surgery simulation using 3D MRI/CT fusion imaging could be useful in determining the indications for full endoscopic surgery for lumbosacral disc herniation.

## Figures and Tables

**Figure 1 medicina-59-00860-f001:**
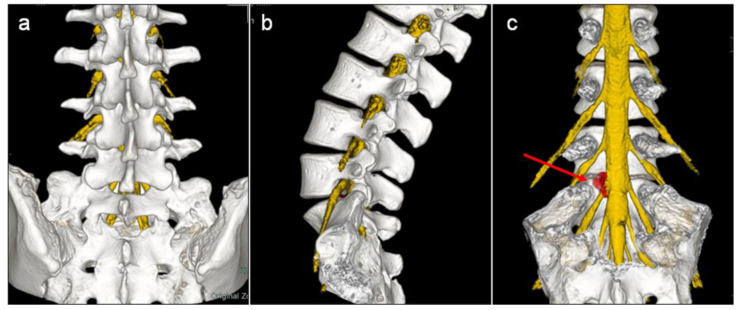
Three-dimensional (3D) fusion images of lumbosacral nerve root-herniated disc (magnetic resonance imaging)–spine-ilium (computed tomography) in the (**a**) postero-anterior view, (**b**) lateral view, and (**c**) postero-anterior view without posterior spinal elements. The red arrow indicates a herniated disc.

**Figure 2 medicina-59-00860-f002:**
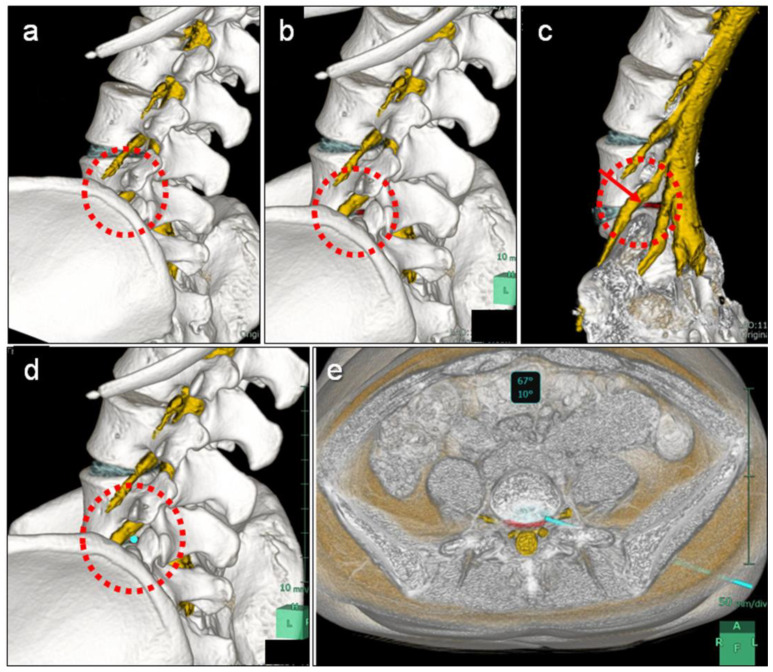
Simulation of full endoscopic lumbar discectomy with the transforaminal approach (FED-TF) at the L5–S1 level using three-dimensional (3D) fusion imaging. The content circled in red is the Kambin’s triangle at the L5–S1 level. The 3D fusion image was rotated along the axial plane of the L5 lower endplate to the (**a**) approach side up to the angle where Kambin’s triangle maximally appears and (**b**,**c**) rotated cephalad so that the ilium did not interfere with Kambin’s triangle when checking whether the herniated disc could be reached. (**d**,**e**) A 3 mm diameter endoscopic virtual axis (light blue bar) was placed according to the FED-TF approach to assess the trajectory.

**Figure 3 medicina-59-00860-f003:**
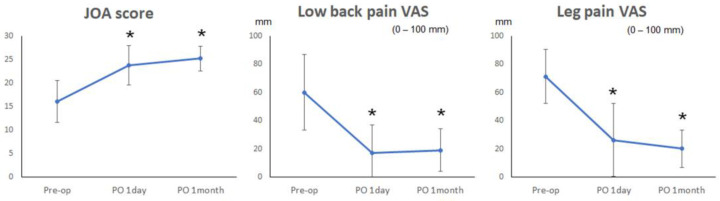
The Japanese Orthopedic Association (JOA) score for lumbar spinal disorders (0–29, with higher scores representing better status) and scores of low back pain and leg pain (visual analog scale [VAS]; 0–100 mm, with higher scores representing worse pain) preoperatively and at 1 day and 1 month postoperatively. Values are in mean ± standard deviation. * *p* < 0.05.

**Figure 4 medicina-59-00860-f004:**
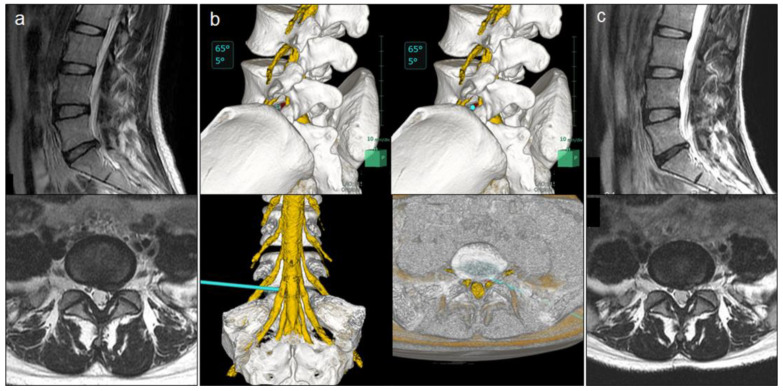
Case presentation of a 28-year-old woman with L5–S1 disc herniation. (**a**) Preoperative magnetic resonance imaging (MRI); (**b**) simulation of full-endoscopic lumbar discectomy with the transforaminal approach using three-dimensional fusion imaging (light blue bar: virtual axis of endoscope); (**c**) postoperative magnetic resonance image.

**Table 1 medicina-59-00860-t001:** Patient demographic data of 52 patients.

	No	Percent
Gender		
Male	28	53.8
Female	24	46.2
Age		
<30 year	10	19.2
31–40 year	12	23.1
41–50 year	15	28.8
>50 year	15	28.8
Disc level		
L5–S1	50	96.2
L5–L6	2	3.8
Classification of herniation		
Protrusion	4	7.7
Subligamentous extrusion	31	59.6
Transligamentous extrusion	14	26.9
Sequestration	3	5.8
Zone of herniated disc		
Central canal zone	13	25.0
Subarticular zone	37	71.2
Foraminal zone	2	3.8
Extraforaminal zone	0	0

## Data Availability

The data that support the findings of this study are available from the corresponding author on reasonable request.

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
