# Peer review of "Evaluation of Surgical Indications for Full Endoscopic Discectomy at Lumbosacral Disc Levels Using Three-Dimensional Magnetic Resonance/Computed Tomography Fusion Images Created with Artificial Intelligence"

_medicina, 2023, doi:10.3390/medicina59050860_

Round 1
Reviewer 1 Report
Dear authors,
Congratulations on your achievement in developing the software program.
Overall, the manuscript was well-written. However, there are a few matters that may need further clarification:
1) In the abstract (line 23), "Thirteen of 52 cases...". It is not clear how the selection was made and what do the 52 cases referring to?
2) I would like to suggest briefly clarifying the purpose of the software program in the abstract.
3) In the introduction (line 64), the authors mentioned: "evaluate the validity of FED-TF surgery". There should be another group of patients that shall be assigned as control, so that can be confirmed on the evaluation part.
4) In the methodology, there was no explanation i.e. inclusion & exclusion criteria of patients. This may help to clarify the sampling issue.
5) Overall, the write-up is most probably can be considered a case-series study rather than an evaluation study.
Author Response
Response to Reviewer #1:
Congratulations on your achievement in developing the software program.
Overall, the manuscript was well-written. However, there are a few matters that may need further clarification:
1) In the abstract (line 23), "Thirteen of 52 cases...". It is not clear how the selection was made and what do the 52 cases referring to?
2) I would like to suggest briefly clarifying the purpose of the software program in the abstract.
Response: Thank you very much for reviewing our manuscript. To clarify the purpose and the cases included in the study, the abstract was revised as follows.
Abstract, line17-
Although fullFull endoscopic lumbar discectomy with the transforaminal approach (FED-TF) is a minimally invasive spinal surgery for lumbar disc herniation, . Although the lumbosacral levels present anatomical challenges when performing FED-TF surgery due to the presence of the iliac bone. In this study, it is difficult to determine whether FED-TF surgery can be safely performed based on preoperative imaging alone. weWe simulated whether FED-TF surgery could be safely performed on a total of 52 consecutive cases with L5–S1 or L5–L6 disc herniation using fused three-dimensional (3D) images of the lumbar nerve root on magnetic resonance imaging (MRI) created with artificial intelligence and lumbosacral spine and iliac on computed tomography (CT) images. Thirteen of 52 cases with L5–S1 or L5–L6 disc herniation were deemed operable according to simulated FED-TF surgery without foraminoplasty using the 3D MRI/CT fusion images.
3) In the introduction (line 64), the authors mentioned: "evaluate the validity of FED-TF surgery". There should be another group of patients that shall be assigned as control, so that can be confirmed on the evaluation part.
Response: The phrase "validity of FED-TF surgery" was not appropriate. The aim of this study is to evaluate the feasibility of FED-TF surgery using preoperative simulation with 3D MRI/CT fusion imaging. We have revised as follows.
Introduction, line 62-
The aim of this study was to analyze the morphology at the L5–S1 level, including that of the iliac crest, during the TF approach using 3D MRI/CT fusion images and to evaluate the feasibility validity of FED-TF surgery for disc herniation using preoperative simulation with 3D MRI/CT fusion imaging for disc herniation.
4) In the methodology, there was no explanation i.e. inclusion & exclusion criteria of patients. This may help to clarify the sampling issue.
Response: The inclusion criteria for this study was patients with a single level of intervertebral disc herniation of L5-S1 or L5-L6 who were deemed to have an indication for surgery. Patients with multi-level herniated discs requiring surgery and patients with spinal canal stenosis requiring decompression surgery were excluded. We have amended the methodology section as follows.
Materials and Methods, line 69-
..., and written informed consent was obtained from all participants. The inclusion criteria was patients with a single level of intervertebral disc herniation of L5-S1 or L5-L6 who were deemed to have an indication for surgery. Patients with multi-level herniated discs and patients with spinal canal stenosis requiring decompression surgery were excluded. Overall, 52 patients...
5) Overall, the write-up is most probably can be considered a case-series study rather than an evaluation study.
Response: As you pointed out, we have amended the introduction section as follows.
Introduction, line 62
The aim of this case-series study was to analyze the morphology at the L5–S1 level, including that of the iliac crest,...

Reviewer 2 Report
Full endoscopic surgery for lumbar disk herniation in an interesting field in evolution. The argument of paper is in interest of general audience.
The paper is interesting because firstly described a simulation method to evaluate the feasibility of approach to lumbosacral junction (L5-S1)
The message of article is clear. I just suggest to short a little the paper with some synthesis. English may be a little improved. Table 2 may be omitted.
Author Response
Response to Reviewer #2
Full endoscopic surgery for lumbar disk herniation in an interesting field in evolution. The argument of paper is in interest of general audience.
The paper is interesting because firstly described a simulation method to evaluate the feasibility of approach to lumbosacral junction (L5-S1)
The message of article is clear. I just suggest to short a little the paper with some synthesis. English may be a little improved. Table 2 may be omitted.
Response: Thank you very much for reviewing our manuscript. As you pointed out, Table 2 was deleted.

Reviewer 3 Report
1. Please clarify/comment if the angles measuered during simulation were used using the actual procedure and if yes , how were the measured during the acutal procedure. How did you ensure the flouro machine is replicating the same angle as the simulation?
2. The devices are available as used by ENT and neurosurgeons where an uploaded images of CT and MRI are used real time during surgery. Any role for that here ?
3. The images were obtained in supine position while surgery was done in the prone position. Did it effect the reliability of the simulation in anyway?
4. How was nerve monitoring done as patient was sedated during the procedure. Did u use any eletrodiagnostic testing intraOp?
5. Was there any discrepency between the simualtion measuremnts and the real time measurement using intraOp flouro ? and how did you resolve it if any existed.
Author Response
Response to Reviewer #3
- Please clarify/comment if the angles measuered during simulation were used using the actual procedure and if yes , how were the measured during the acutal procedure. How did you ensure the flouro machine is replicating the same angle as the simulation?
Response: Thank you very much for reviewing our manuscript. In the actual surgery, the endoscope was inserted using the puncture point and angle measured during the simulation. The arm of the fluoro machine was aligned with the axial direction of the endoscope, and the angle meter attached to the device was used to confirm that the arm angle was the same as in the simulation. However, the actual insertion angle of the endoscope may not be accurate due to misalignment of body angles. Therefore, we are analyzing the actual endoscopic insertion route from intraoperative fluoroscopic images in three dimensions and examining the accuracy of the preoperative simulation route in the future. We have amended the surgical procedure section and discussion section as follows.
Materials and Methods, Surgical procedure, line 137-
A standard transforaminal full endoscopic lumbar discectomy (inside-out technique) was performed without partial osteotomy of the SAP (foraminoplasty) [6,11,21]. The endoscope was inserted using the puncture point and angle measured during the simulation. The arm of the fluoro machine was aligned with the axial direction of the endoscope, and the angle meter attached to the device was used to confirm that the arm angle was the same as in the simulation. Surgery was performed...
Discussion, line 257-
...the margin of the SAP to be resected upon approaching the herniated disc. We did not compare the actual and the simulated surgical pathways in this study, which is another limitation of this study. Regarding this issue, we are analyzing the actual endoscopic insertion route from intraoperative fluoroscopic images in three dimensions and examining the accuracy of the preoperative simulation route in the future. The low number of cases is also a limitation.
- The devices are available as used by ENT and neurosurgeons where an uploaded images of CT and MRI are used real time during surgery. Any role for that here ?
Response: In the future, we believe that this surgical simulation using 3D MRI/CT fusion imaging has the potential to be applied to the navigation system that can be used in real time during surgery. We have added the following to the Discussion section.
Discussion, line 230
...the insertion angle and path of the endoscope from multiple angles and puncture points from the body surface (Figure 2). In the future, we believe that this surgical simulation using 3D MRI/CT fusion imaging has the potential to be applied to the navigation system that can be used in real time during surgery.
- The images were obtained in supine position while surgery was done in the prone position. Did it effect the reliability of the simulation in anyway?
Response: We have mentioned the issue of body position as a limitation (line 248-251), and as you pointed out, this 3D image-based surgical simulation may be affected by body position. Since the actual safety zone may be enlarged during surgery in the prone position compared to the preoperative simulation as previously reported, the indications for FED-TF surgery may be expanded beyond the indications in this simulation. We have amended the limitation section as follows.
Discussion, line 248
First, CT and MR imaging were performed in the supine position, which is different from the pronesupine position during surgery. The actual safety zone may be enlarged during surgery in the pronesupine position compared to the preoperative simulation as previously reported [10,11], and the indications for FED-TF surgery may be expanded beyond the indications in this simulation. Second,...
- How was nerve monitoring done as patient was sedated during the procedure. Did u use any eletrodiagnostic testing intraOp?
Response: The surgery is performed under local anesthesia with sedation used only enough to make the patient responsive to mild stimulation. Therefore, electrodiagnostic testing system are not used because it is possible to detect whether nerve roots have been stimulated during the surgery. We have amended the materials and methods section.
Materials and Methods, Surgical procedure, line 138-
Surgery was performed under local anesthesia with sedation used only enough to make the patient responsive to mild stimulation administered by anesthesiologists [22]. Electrodiagnostic testing system are not used because it is possible to detect whether nerve roots have been stimulated during the surgery. The patient positioned prone...
- Was there any discrepency between the simualtion measuremnts and the real time measurement using intraOp flouro ? and how did you resolve it if any existed.
Response: As we answered in Q1, the endoscope was inserted using the puncture point and angle measured during the simulation. However, we did not compare the actual surgical pathway and insertion angle with the simulated one in this study, which we add as a limitation of this study.
Discussion, line 257-
...the margin of the SAP to be resected upon approaching the herniated disc. We did not compare the actual and the simulated surgical pathways in this study, which is another limitation of this study. Regarding this issue, we are analyzing the actual endoscopic insertion route from intraoperative fluoroscopic images in three dimensions and examining the accuracy of the preoperative simulation route in the future. The low number of cases is also a limitation.
